# Inhibition of HDAC8 Reduces the Proliferation of Adult Neural Stem Cells in the Subventricular Zone

**DOI:** 10.3390/ijms25052540

**Published:** 2024-02-22

**Authors:** Momoko Fukuda, Yuki Fujita, Yuko Hino, Mitsuyoshi Nakao, Katsuhiko Shirahige, Toshihide Yamashita

**Affiliations:** 1Department of Anatomy and Developmental Biology, School of Medicine, Shimane University, 89-1, Enya-cho, Izumo-shi 693-8501, Japan; 2Institute of Molecular Embryology and Genetics, Kumamoto University, 2-2-1 Honjo, Chuo-ku 860-0811, Japan; 3Laboratory of Genome Structure and Function, Institute for Quantitative Biosciences, The University of Tokyo, 1-1-1 Yayoi, Bunkyo-Ku, Tokyo 113-0032, Japan; 4Department of Cell and Molecular Biology, Karolinska Institutet, Biomedicum, Quarter A6, 171 77 Stockholm, Sweden; 5Department of Molecular Neuroscience, Graduate School of Medicine, Osaka University, 2-2, Yamadaoka, Suita 565-0871, Japan; 6WPI Immunology Frontier Research Center, Osaka University, 3-1, Yamadaoka, Suita 565-0871, Japan; 7Graduate School of Frontier Biosciences, Osaka University, 1-3, Yamadaoka, Suita 565-0871, Japan; 8Department of Neuro-Medical Science, Graduate School of Medicine, Osaka University, 2-2, Yamadaoka, Suita 565-0871, Japan

**Keywords:** adult neurogenesis, neural stem cells, histone deacetylase 8 (HDAC8), cytokine-mediated signaling, subventricular zone

## Abstract

In the adult mammalian brain, neurons are produced from neural stem cells (NSCs) residing in two niches—the subventricular zone (SVZ), which forms the lining of the lateral ventricles, and the subgranular zone in the hippocampus. Epigenetic mechanisms contribute to maintaining distinct cell fates by suppressing gene expression that is required for deciding alternate cell fates. Several histone deacetylase (HDAC) inhibitors can affect adult neurogenesis in vivo. However, data regarding the role of specific HDACs in cell fate decisions remain limited. Herein, we demonstrate that HDAC8 participates in the regulation of the proliferation and differentiation of NSCs/neural progenitor cells (NPCs) in the adult mouse SVZ. Specific knockout of *Hdac8* in NSCs/NPCs inhibited proliferation and neural differentiation. Treatment with the selective HDAC8 inhibitor PCI-34051 reduced the neurosphere size in cultures from the SVZ of adult mice. Further transcriptional datasets revealed that HDAC8 inhibition in adult SVZ cells disturbs biological processes, transcription factor networks, and key regulatory pathways. HDAC8 inhibition in adult SVZ neurospheres upregulated the cytokine-mediated signaling and downregulated the cell cycle pathway. In conclusion, HDAC8 participates in the regulation of in vivo proliferation and differentiation of NSCs/NPCs in the adult SVZ, which provides insights into the underlying molecular mechanisms.

## 1. Introduction

Adult neurogenesis is the process of generating functional neurons from adult neural stem cells (NSCs), which can differentiate into both neurons and glia that compose the central nervous system and replicate through cell division [1].

NSCs are present in adults, and neurogenesis occurs throughout an individual’s lifetime. In the adult mammalian brain, NSCs are mainly present in limited locations; one of the most important areas is the subventricular zone (SVZ). In the SVZ, neural stem cells (B cells) produce transient amplifying cells (C cells) that are intermediate neural progenitors. These transient amplifying cells undergo repeated division in a short cell cycle and produce numerous immature neurons. Immature neurons (A cells) pass through the rostral migratory stream within the astrocytes and migrate to the olfactory bulb to differentiate and mature. Each stage of differentiation is controlled by changes in gene expression.

At each stage of neurogenesis, epigenetic modifications, such as histone deacetylation, could provide a coordinated system for regulating gene expression. Histone deacetylation is a common epigenetic modification of the chromatin associated with gene silencing, resulting in the suppression of gene expression due to increased chromatin compression and reduced accessibility to transcription factors. Mammalian histone deacetylases (HDACs) comprise a superfamily of four classes based on domain organization [2,3]. HDAC1, HDAC2, HDAC3, and HDAC8 are class I members that are ubiquitously expressed with predominant nuclear localization.

The functions in adult neurogenesis of all class 1 HDACs, except for HDAC8, have been reported [4,5,6]. However, HDAC8 expression in adult NSCs/neural progenitor cells (NPCs) has not been fully addressed, and the specific role of HDAC8 in neurogenesis remains elusive. HDAC8 expression has been detected using a serial analysis of gene expression in brain tumor tissue. Moreover, increased expression of HDAC8 is associated with poor outcomes in neuroblastoma [7,8]. Furthermore, inhibiting HDAC8 with selective HDAC8 inhibitors or siRNA-mediated knockdown reduces cell proliferation in cultured neuroblastoma cell lines and in vivo using xenograft mouse models [9]. These findings suggest that HDAC8 regulates proliferation in the pathological brain. In the physiological context, NSCs/NPCs can proliferate in the adult brain. In this study, we assessed the possible regulatory role of HDAC8 in the physiological proliferation of NSCs/NPCs in the adult brain.

## 2. Results

### 2.1. HDAC8 Expression in the Adult SVZ

To investigate the role of HDAC8 in adult neurogenesis, we first examined its expression pattern in the SVZ. HDAC8 was highly expressed in Nestin-positive neural stem/progenitor cells (Figure 1a). The additional small pictures show the magnified images of the dorsolateral corner of SVZ. The cells in the SVZ are classified into three major types [1], namely radial glia-like cells (GFAP-positive cells) [10], transient amplifying cells (MASH1-positive cells) [11], and neuroblast cells (doublecortin (Dcx) [12,13,14,15,16]. Different antibodies (anti-GFAP, anti-MASH1, and anti-Dcx) were used as co-labels with HDAC8 to detect the types of NSCs/NPCs (Figure 1b). These findings suggest the involvement of HDAC8 in neurogenesis in the adult brain.

### 2.2. HDAC8 Deficiency Inhibits Adult Neurogenesis

To assess the function of HDAC8 in adult neurogenesis in vivo, *Hdac8* was specifically deleted in Nestin-positive neural stem/progenitor cells. *Hdac8 flox/flox* mice [17] were bred with Nestin-CreERT2 mice [18] that express tamoxifen-inducible Cre recombinase under the control of the Nestin promoter and enhancer. Figure 2a shows the timeline of tamoxifen administration in adult Nestin-CreERT2; *Hdac8 flox/flox* mice. Specific deletion of *Hdac8* in the SVZ was confirmed by the loss of immunoreactivity for HDAC8 (Figure 2b). We conducted immunofluorescence staining of the adult brain sections using antibodies against neural stem/progenitor cells or neuroblast markers to investigate the effects of *Hdac8* deletion on SVZ cells.

We observed reduced expression of GFAP, Nestin, and DCX in tamoxifen-treated Nestin CreERT2; *Hdac8 flox/flox* mice compared with that in *Hdac8 flox/flox* mice (Figure 2c,d). In contrast, no significant differences were observed for MASH1. We did not detect the HDAC8 expression in the MASH1-positive cells (Figure 2e). These observations suggest that *Hdac8* deletion reduces the number of NSCs/NPCs and inhibits adult neurogenesis in the SVZ.

### 2.3. Deletion of Hdac8 Suppresses the Proliferation of Adult NSCs/NPCs

We further investigated the effect of *Hdac8* deletion on adult NSC proliferation. The brain sections were immunohistochemically processed to analyze the expression of the cell proliferation marker Ki67 [19]. The number of Ki67-positive cells decreased in the SVZ of tamoxifen-treated Nestin CreERT2; *Hdac8 flox/flox* mice compared with that observed in *Hdac8 flox/flox* mice (Figure 3a,b). These findings suggest that the deletion of *Hdac8* inhibits the proliferation of adult NSCs/NPCs.

Cells isolated from the adult SVZ can proliferate in a medium containing specific growth factors to produce multipotent clonal aggregates called neurospheres [20,21]. To investigate the cell-intrinsic functions of HDAC8, we further examined the role of HDAC8 inhibition in cultured neurospheres prepared from the adult SVZ. Treating the SVZ cells with the selective HDAC8 inhibitor PCI-34051 reduced the diameter of neurospheres compared with vehicle-treated control cells (Figure 4). The negative effect of PCI-34051 on the proliferation of neurospheres was detected in a dose-dependent manner (Figure 4c). To further assess NPC proliferation, we measured the EdU incorporation of neurospheres. EdU was administered 30 min before fixation of the neurospheres. The inhibition of HDAC8 resulted in a significant reduction in EdU incorporation (Figure 4d,e). These observations suggest that HDAC8 participates in the regulation of the proliferation of adult SVZ cells.

### 2.4. Expression Analysis Reveals HDAC8 Inhibition Contributes to Inhibition of the Cell Cycle

HDAC8 is an epigenetic player that is linked to deregulated gene expression. To characterize the transcriptomic features in response to HDAC8 inhibitor treatment in the neurospheres derived from adult SVZs, we performed an RNA-seq analysis. We detected differentially expressed genes (DEGs) from the RNA-seq dataset using RNAseqChef [22]. We found reliable in vitro datasets of cultured neurospheres, which were treated with vehicle (Control) or 20 μM of HDAC8 inhibitors. Principal component analysis (PCA) showed that the datasets from the Control or HDAC8 inhibitor were distinctly positioned (Figure 5a). We detected DEGs in the HDAC8 inhibitor-treated group compared to the control group; HDAC8 inhibitor treatment upregulated 105 genes and downregulated 90 genes (Figure 5b). To clarify the biological signaling pathway related to HDAC8 inhibition, we performed enrichment analysis based on the KEGG gene set. The top-ranked pathways related to the 105 upregulated genes were the cytokine–cytokine receptor interaction, nod-like receptor signaling pathway, and chemokine signaling pathway (Figure 5c–e). Enriched terms included cytokine–cytokine receptor interaction and chemokine signaling pathway and included genes such as *Ccl* and *Cxcl* (Figure 5f–h).

We also analyzed downregulated genes in HDAC8 inhibitor-treated neurospheres using enrichment analysis. Enrichment analysis showed that the 90 downregulated genes in adult SVZ neurospheres after HDAC8 inhibition enriched the cell cycle pathway, such as *Cdk1* and *Cdc20* (Figure 5f,g,i). These gene expression profiles suggest cell cycle arrest following HDAC8 inhibitor treatment.

## 3. Discussion

In this study, we evaluated the role of HDAC8 in adult NSCs/NPCs using Nestin-CreERT2 mice and demonstrated that the loss of HDAC8 in adult Nestin-positive cells reduces NSCs/NPCs and neuroblasts in the SVZ.

During adult neurogenesis, NSCs and progenitors show various gene expression patterns that are progressively altered as they commit to a neurogenic lineage in the SVZ and dentate gyrus. Therefore, evaluation of the epigenetic mechanisms controlling various gene expression patterns is important to regulate neurogenesis in the adult brain [23]. For instance, in mice lacking the catalytic activity of HDAC2 or with a conditional deletion of HDAC2, neurons derived from adult neurogenesis undergo cell death at a specific maturation stage, suggesting that HDAC2 plays a critical role in silencing the transcripts related to cell death during neuronal differentiation, though the detailed downstream signaling remains elusive [4].

Reports on HDAC8 expression in adult NSCs/NPCs are lacking. In this study, we detected high HDAC8 expression in Nestin-positive cells in adult SVZ, suggesting the role of HDAC8 in adult neurogenesis. Moreover, we showed that HDAC8 inhibition caused a reduction in the size of SVZ neurospheres, suggesting that HDAC8 regulates the proliferation of NSCs/NPCs in the adult mouse SVZ. In contrast, loss of *Hdac8* did not affect the number of MASH1-positive transit-amplifying cells, which underwent active proliferation. However, *Hdac8* deletion decreased DCX; this is possibly because of the delayed commitment of MASH1-positive transit-amplifying cells to become DCX-positive neuroblasts, which may result in countervailing reduced proliferation.

We also attempted to elucidate the molecular mechanism underlying the regulation of the proliferation of adult NSCs/NPCs by HDAC8. HDAC8 shares low sequence similarity with other class I HDACs and is most closely similar to HDAC3, with only 34% identity [7]. Although HDAC8 differs from other class I HDACs in sequence identity, it contains a deacetylase catalytic domain [24,25]. As with other HDACs, HDAC8 can induce deacetylation of both histone [26,27] and nonhistone proteins [28,29,30,31,32] in vitro. Therefore, HDAC8 may regulate adult neurogenesis through a mechanism mediated by nonhistone substrates rather than epigenetic modification by histone deacetylation.

The HDAC8 inhibitor PCI-34051, developed through modification of a low molecular weight hydroxamic acid scaffold, demonstrated promising potency and selectivity against HDAC8 compared with other class I HDACs. PCI-34051 inhibits recombinant HDAC8 with a Ki of 10 nM with >200-fold selectivity over the other HDACs [33,34]. To determine the functional consequences in neurospheres treated with an HDAC8 inhibitor, we performed RNA-seq, followed by enrichment analysis of the differentially expressed genes using RNAseqChef, and detected upregulation of chemokines and downregulation of cell cycle-related genes. Our findings indicated that PCI-34051 delayed cell proliferation. Further studies, such as RNA-seq analysis siRNA-mediated knockdown of HDAC8 or knockout of HDAC8, will help to address the precise effect of HDAC8-selective inhibition.

Causal variants of HDAC8 have been reported in individuals with Cornelia de Lange Syndrome (CdLS) and in a family with X-linked intellectual disability [29,35,36,37,38,39]. Intellectual disability, well-defined facial features, and upper limb anomalies characterize CdLS [40]. In addition to the variants in NIPBL, which is required for cohesin complex loading onto chromatin, and cohesin core, its regulatory proteins have been reported as the cause of CdLS, including HDAC8 [41]. The HDAC8 mutation in CdLS is associated with HDAC8 dysfunction and results in increased acetylation of Smc3, one of the subunits of the cohesin complex [29]. Therefore, deacetylation of Smc3 by HDAC8 might be associated with the development of normal brain functions. It should be noted that the CdLS phenotypes are governed by the developmental loss of HDAC8 functions. In our study, deletion of HDAC8 in the adult stage also caused neurogenesis deficits, suggesting the potential involvement of HDAC8 in maintaining neuronal function.

In conclusion, our findings indicated that the loss of HDAC8 affected physiological neurogenesis, which could be useful in assessing the adverse effects of an HDAC8 inhibitor that is being developed as a therapeutic drug and the physiological functions of HDAC8 in the central nervous system. Our findings prompt us to investigate the potential applications of PCI-34051 in diseases of the nervous system, such as neuroblastoma. Further studies identifying the HDAC8 target would facilitate the demonstration of the effectiveness of an HDAC-selective inhibitor in the clinical model. Structural specificities of HDAC8 have allowed for the design of selective inhibitors [42], and isotype-specific targeting for HDAC8 could be an attractive strategy. Elucidating the physiological functions of HDAC8 would help to understand mechanisms in neurological diseases and develop pharmacological interventions.

## 4. Materials and Methods

### 4.1. Experimental Animals

C57BL/6J mice obtained from Japan SLC, Inc. (Shizuoka, Japan) were bred and maintained at the Institute of Experimental Animal Sciences, Osaka University Graduate School of Medicine, Osaka, Japan. Conditional *Hdac8 flox* mice were provided by Dr. Eric N. Olson (The University of Texas, Austin, TX, USA) [17]. Nestin-CreERT2 mice were provided by Dr. Ryoichiro Kageyama and Dr. Itaru Imayoshi (Institute for Virus Research, Kyoto University, Kyoto, Japan) [18].

To delete *Hdac8* in adult Nestin-positive cells, tamoxifen (10 mg/35 g of mouse) was orally administered in mice once a day for 4 sequential days at postnatal (P) weeks 8 and 12 [43].

### 4.2. Immunohistochemistry

The mice underwent transcardial perfusion with PBS, followed by 4% paraformaldehyde in phosphate buffer (0.1 M). Subsequently, the brains were dissected, postfixed in the same fixative, immersed overnight in PBS containing 30% sucrose, embedded in Tissue-Tek OCT, and frozen at −80 °C until further use. Brain sections were prepared using a cryostat (20-µm thickness) and mounted on Matsunami adhesive-coated slides (Matsunami, Osaka, Japan). Cryostat sections were incubated with blocking solution containing 5% BSA and 0.1% Triton X-100 in PBS for 1 h at room temperature, followed by overnight incubation at 4 °C with primary antibodies including anti-HDAC8 and DCX (Santa Cruz Biotechnology, Inc., Santa Cruz, CA, USA), anti-Nestin (Merck Millipore, Burlington, MA, USA), anti-GFAP (Sigma-Aldrich, St. Louis, MO, USA or Dako, Glostrup, Denmark), and anti-MASH1 and anti-Ki67 (BD Pharmingen, San Jose, CA, USA). For mouse primary antibody, Vector^®^ M.O.M.™ immunodetection Kit (Vector Laboratories, Burlingame, CA, USA) was used as per the manufacturer’s protocol. Immunoreactivity was visualized using Alexa Flour 488- or 568-conjugated secondary antibodies (Thermo Fischer Scientific, Rockford, IL, USA). Coverslips were placed on the slides with mounting medium (Dako). Nuclei were stained using 4′, 6-diamidino-2-phenylindole (DAPI). Images were captured using a laser scanning confocal microscope (FV3000, Olympus, Tokyo, Japan).

### 4.3. Quantification of SVZ Cells

The number of cells in the SVZ was determined through serial section analysis. Immunohistochemistry was performed as described above. Every tenth coronal section (20-μm thickness) along the rostral–caudal axis of the SVZ was analyzed. The percentage of positive area indicated by antibody staining was measured using Image J software 64-bit (NIH, Bethesda, MD, USA). Each area was normalized to the SVZ area detected by DAPI-staining of the same sections.

### 4.4. Neurosphere Culture

The neurospheres were cultured following a previously described procedure [44]. For each preparation, the mouse brains were dissected to isolate SVZ; the SVZ tissues were minced in ice-cold HBSS containing glucose (30 mM), HEPES (2 mM), and NaHCO_3_ (26 mM). Digestion was performed in 0.05% Trypsin-EDTA for 20 min; the reaction was terminated by adding an equal volume of trypsin inhibitor in DPBS (T6522, Sigma-Aldrich, Darmstadt, Germany) and incubated for 20 min. The digested tissue was triturated to obtain single cells using pipettes, and cells were pelleted at 200× *g* for 5 min. Cells were washed three times in proliferation medium (Neural Stem Cell Basal Medium, SCM003, Merck Milipore, Darmstadt, Germany), supplemented with B27 without vitamin A (12587010, Thermo Fisher Scientific, MA, USA), GlutaMAX (35050061, Thermo Fisher Scientific, MA, USA), Antibiotic-Antimycotic (15240062, Thermo Fisher Scientific, Waltham, MA, USA), epidermal growth factor (EGF) (1 µg/mL) (E9644, Sigma-Aldrich, Germany), and basic fibroblast growth factor (FGF2) (1 µg/mL) (100-18B, PeproTech, Rocky Hill, NJ, USA). Cells were plated into a single well of a 24-well plate. Half of the proliferation medium was continuously changed every other day. The cells isolated from SVZ are positive for several well-characterized NSC markers, such as Nestin and Sox2, and can proliferate, as demonstrated using a BrdU incorporation assay.

### 4.5. Neurosphere Assay

Adult SVZ cells were seeded into 24-well culture plates at 1 × 10^5^ cells/mL in proliferation media treated with the selective HDAC8 inhibitor PCI-34051 (Cayman Chemical Company, Ann Arbor, MI, USA) or vehicle control (Dimethyl sulfoxide, DMSO). Neurospheres were allowed to develop for 7 days in an incubator with 5% CO_2_ in a humidified atmosphere at 37 °C. The average neurosphere diameter was assessed in three 2 mm^2^ squares in each well.

To assess the proliferation of neurospheres derived from adult SVZs in vitro, neurospheres were plated on chamber slide glasses, incubated for 30 min, and then exposed to EdU (10 μM) for another 30 min at 37 °C, followed by fixation in 4% paraformaldehyde. To detect incorporated EdU (Invitrogen, Waltham, MA, USA, E10187), we used a Click-iT^®^ EdU Imaging Kit (Thermo Fisher Scientific, MA, USA) following the manufacturer’s instructions.

### 4.6. RNA Isolation and RNAseq Analysis

The RNA isolation was performed using a previously described method [45]. Briefly, total RNA was isolated using Trizol (Invitrogen, MA, USA), and 5 μg of total RNA was purified using the RNeasy Mini Kit (QIAGEN, Hilden, Germany). Furthermore, 100–200 ng of mRNA was fragmented through hydrolysis and purified (RNAeasy Minelute Kit; QIAGEN, Germany). Library DNAs were prepared according to the Illumina TrueSeq protocol using the Truseq Standard mRNA LT Sample Prep Kit (Illumina, San Diego, CA, USA) and sequenced by Illumina NextSeq 500 (Illumina, CA, USA) using the Nextseq 500/550 High Output v2.5 Kit (Illumina, CA, USA) to obtain single-end 75 bp reads. The resulting reads were aligned to the mouse genome (mm10) using STAR ver.2.6.0a after trimming to remove the adapter sequence and low-quality ends using Trim Galore! v0.5.0 (cutadapt v1.16). The transcript abundance was determined using RSEM v1.3.1. The counts of gene-expression profiles in those selected datasets were further analyzed through RNAseqChef web-based transcriptome analysis [22]. Pathway Interaction Database was used for enrichment analysis in RNAseqChef. The Z-scored normalized count is shown in the heatmaps (genefilter: methods for filtering genes from high-throughput experiments. R package version 1.72.1.).

### 4.7. Statistical Analysis

Statistical analyses were performed using GraphPad Prism 7 (GraphPad Software, San Diego, CA, USA). Quantitative data are expressed as the mean ± standard error of at least three independent experiments. Differences between pairs of experimental groups were analyzed using the Student’s *t*-test or one-way ANOVA, followed by Tukey’s multiple comparisons test. Statistical significance was defined at values of *p* < 0.05.

## Figures and Tables

**Figure 1 ijms-25-02540-f001:**
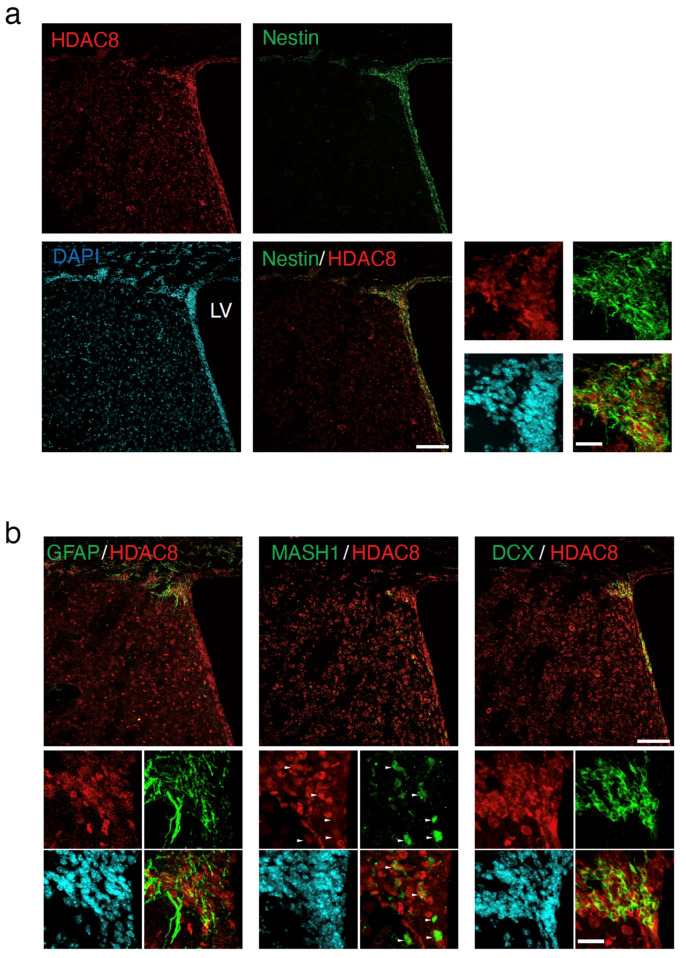
Expression of HDAC8 in the adult mice SVZ. (**a**) The expression of HDAC8 (red) in the Nestin-positive cells (green) in the adult mice subventricular zone (SVZ). Nuclei were counterstained with DAPI (blue). The additional small pictures show the dorsolateral corner of SVZ. LV, lateral ventricle. (**b**) The immunofluorescence signal for HDAC8 (red) was detected in the cells co-immunostained with anti-GFAP, anti-MASH1, and anti-doublecortin (DCX) antibodies. The additional small pictures show the dorsolateral corner of SVZ. White arrowheads indicate the representative location of MASH1-positive cells in the SVZ co-immunostained with anti-MASH1 and anti-HDAC8 antibodies. Scale bars = 100 µm in low magnification image (large panels) and 20 µm in high magnification image (small panels).

**Figure 2 ijms-25-02540-f002:**
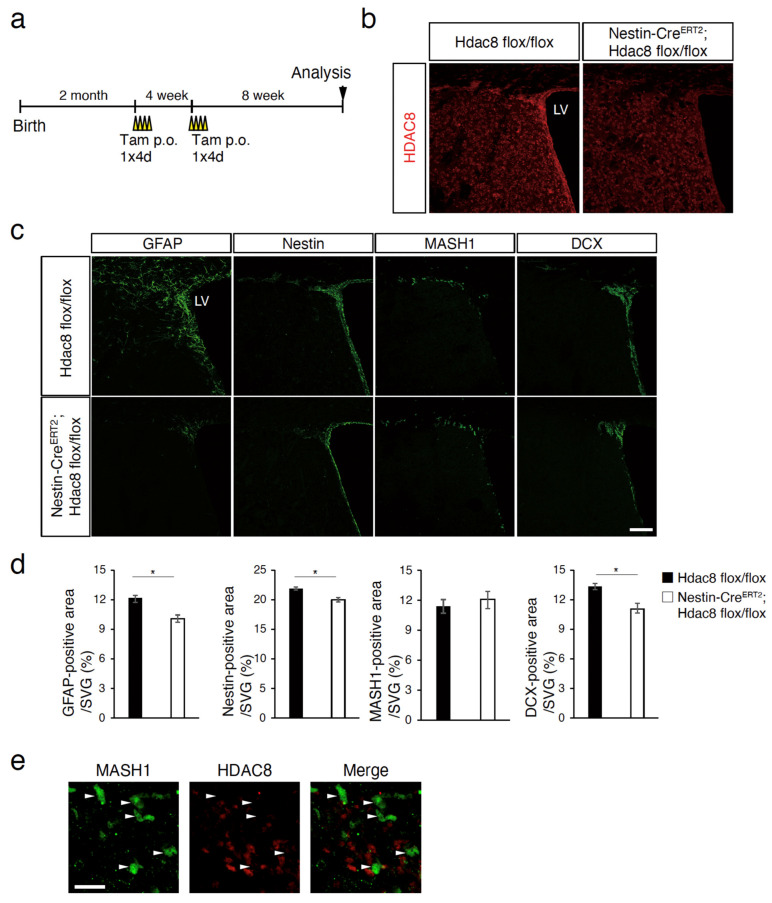
Loss of HDAC8 reduces neurogenesis in adult mice SVZ. (**a**) Timeline of treatment with tamoxifen. p.o.: per os. (**b**) Deletion of HDAC8 in the subventricular zone (SVZ) of tamoxifen-treated Nestin-CreERT2; *Hdac8 flox/flox* adult mice. The coronal brain sections of the adult mice SVZ were immunostained with an anti-HDAC8 antibody (red). LV: lateral ventricle. (**c**) The brain sections that were immunostained with anti-GFAP, anti-Nestin, anti-MASH1, and anti-doublecortin (anti-DCX) antibodies. (**d**) Graphs showing the percent areas of indicated cells normalized to the SVZ area of tamoxifen-treated *Hdac8 flox/flox* or Nestin-CreERT2; *Hdac8 flox/flox* mice. GFAP, *n* = 5; Nestin, *n* = 3; MASH1, *n* = 3; DCX, *n* = 5. * *p* < 0.05, Student’s *t*-test. (**e**) The images show the dorsolateral corner of SVZ stained with MASH1 and HDAC8. White arrowheads indicate the representative location of MASH1-positive cells in the SVZ co-immunostained with anti-MASH1 and anti-HDAC8 antibodies. Scale bars = 100 µm (**b**,**c**) and 20 µm (**e**).

**Figure 3 ijms-25-02540-f003:**
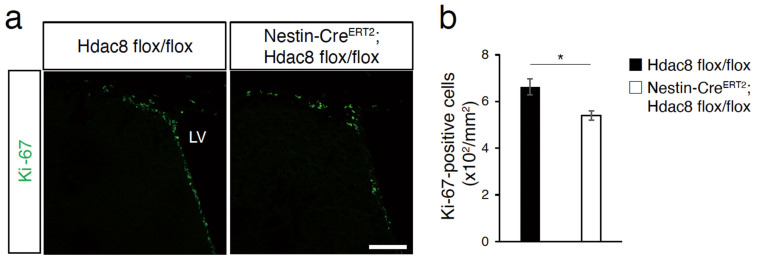
Deletion of HDAC8 reduces the proliferation of adult neural stem cells in the SVZ. (**a**) Distribution of Ki67 (green) in the SVZ. (**b**) The numbers of Ki67-positive cells in the SVZ decreased in the tamoxifen-treated Nestin-CreERT2; *Hdac8 flox/flox* mice compared with those observed in Hdac8 flox/flox mice. *n* = 4. * *p* < 0.05, Student’s *t*-test. Scale bar, 100 µm.

**Figure 4 ijms-25-02540-f004:**
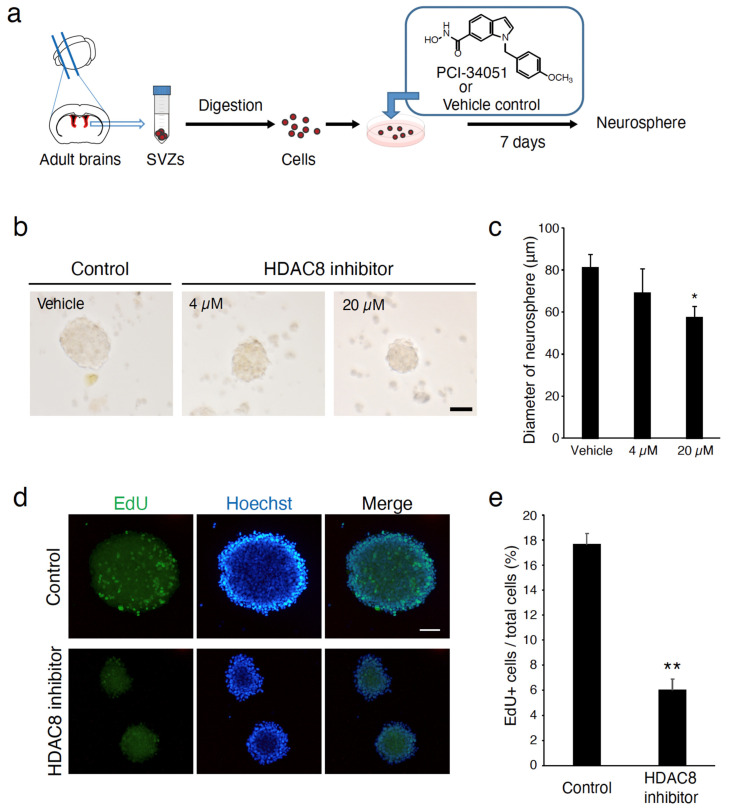
Deletion of HDAC8 reduces the diameter of neurospheres in the SVZ. (**a**) Schematic presentation of the preparation of SVZ neurospheres. (**b**) Cultured adult SVZ neurospheres in the presence of a selective HDAC8 inhibitor, PCI-34051. Scale bars = 50 µm. (**c**) NSCs/NPCs produced in adult SVZ were seeded and treated with PCI-34051 or vehicle control (DMSO), and the average neurosphere diameters were assessed. *n* = 5, * *p* < 0.05, one-way ANOVA followed by Tukey’s multiple comparisons test. (**d**) Representative confocal images of EdU+ cells in vehicle control (DMSO) or HDAC8 inhibitor-treated neurospheres. EdU was administered 30 min before the fixation of neurospheres. Nuclei were stained with Hoechst. Scale bars = 50 µm. (**e**) EdU incorporation into the neurospheres was significantly reduced by treatment with HDAC8 inhibitors. *n* = 5. ** *p* < 0.01, Student’s *t*-test.

**Figure 5 ijms-25-02540-f005:**
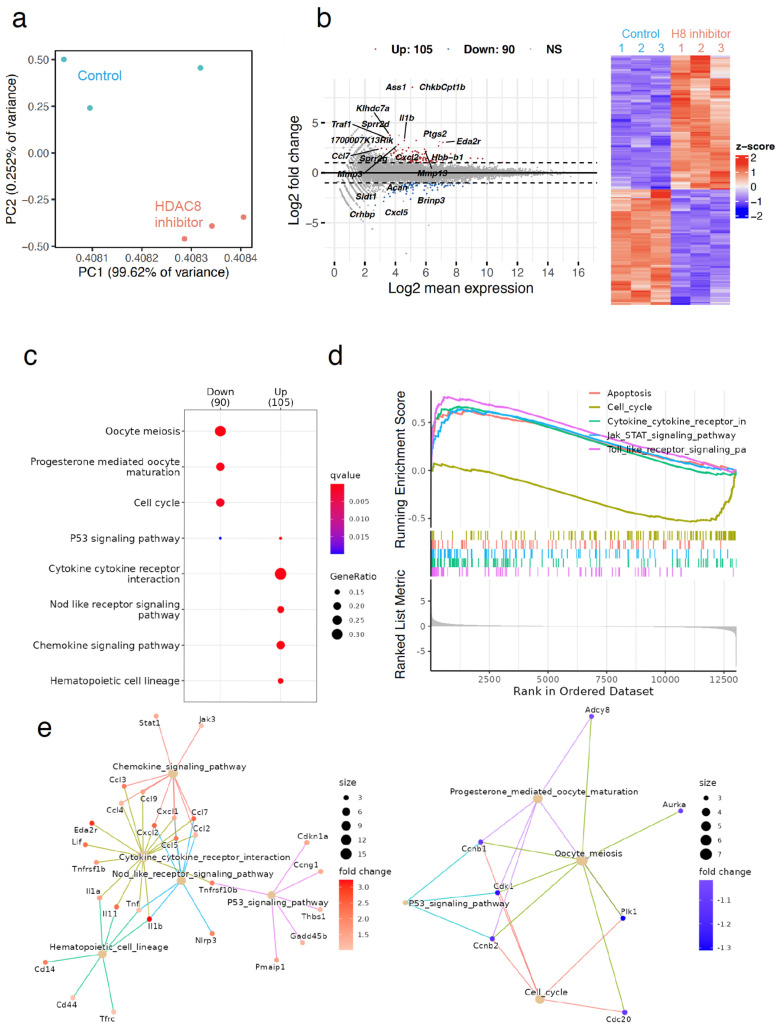
Inhibition of HDAC8 induces transcriptomic changes in neurospheres derived from adult SVZ. (**a**) Transcriptome-based PCA of Control (vehicle treatment) and HDAC8 inhibitor-treated neurospheres. (**b**) Gene expression profiles in Control (vehicle treatment) and HDAC8 inhibitor-treated neurospheres were determined using RNA-seq and subjected to differentially expressed gene (DEG) analysis using the RNAseqChef web tool. The results of the DEG analysis are shown in an MA plot and heatmap. (**c**–**e**) Top-ranked signaling pathways were enriched in gene sets altered by HDAC8 inhibitor treatment, including 105 upregulated genes and 90 downregulated genes. Enrichment analysis was performed based on the KEGG gene set. FDR < 0.05. (**f**–**i**) The genes involved in the top-ranked signaling pathways were indicated in the volcano plot (**f**). Normalized expression changes of the representatives for upregulated (**h**) or downregulated (**i**) by HDAC8 inhibitor treatment. * *p* < 0.05, ** *p* < 0.01, Welch’s *t*-test. FDR, false discovery rate; PCA, Principal component analysis.

## Data Availability

The data presented in this study are available on request from the corresponding author.

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
