# Peer review of "Inhibition of HDAC8 Reduces the Proliferation of Adult Neural Stem Cells in the Subventricular Zone"

_ijms, 2024, doi:10.3390/ijms25052540_

Round 1

Reviewer 1 Report

Comments and Suggestions for Authors

In their manuscript entitled "Inhibition of HDAC8 reduces the proliferation of adult neural stem cells in the subventricular zone," Fukuda et al. explore the impact of Histone deacetylase 8 (HDAC8) inhibition on neuronal stem cells (NSCs) from the subventricular zone (SVZ) in the adult mouse brain. The study sheds light on the epigenetically regulated neurogenesis, presenting well-designed and executed experiments.

The manuscript is written in clear and understandable English, appropriately illustrated, and explained. However, there are a few critical and minor remarks to consider.

Major Remarks:

Clarification of Marker Choices:

The International Journal of Molecular Sciences caters to a multidisciplinary audience, necessitating a brief explanation in the Results section about the functions of GFAP, Mash1, and Dcx, along with justification for their selection as markers.

Precision in Statements:

Certain expressions in the Abstract and Results section, implying direct regulation by HDAC8, lack firm evidence. Suggested revisions include using terms like "participates in the regulation" instead of "regulates" to accurately reflect the findings.

Minor Remarks:

Figure Descriptions:

In Figure 1a, the additional small pictures lack explanation, requiring clarification in both figure captions and the text. Additionally, Figure 3e lacks accompanying information.

Scale Bar Information:

Figures 1, 2, and 3 lack information about the size of scale bars. While Figure 4 mentions a scale bar of "50 mm," this seems implausible and needs correction.

Figure Details:

The meaning of white arrowheads/triangles in some pictures remains unclear and should be explained for reader understanding.

Word Choice:

The term "stimulated" at line 139 might be misleading when referring to the action of an inhibitor; consider a more appropriate term.

Overall Assessment:

Despite these discrepancies, Fukuda et al.'s work is commendable and presents a valuable scientific contribution. With the suggested revisions, the manuscript can be accepted for publishing in IJMS.

Author Response

I wish to re-submit the manuscript titled “Inhibition of HDAC8 reduces the proliferation of adult neural stem cells in the subventricular zone.”  

We thank the Reviewer for your thoughtful suggestions and insights. The manuscript has been rechecked and the necessary changes have been made in accordance with the Reviewer's suggestions. The responses to all comments have been prepared and attached.  

Reviewer 2 Report

Comments and Suggestions for Authors

The manuscript entitled „Inhibition of HDAC8 reduces the proliferation of adult neural stem cells in the subventricular zone” reveals an interesting role of HDAC8 in regulation of NSCs in the adult SVZ. The subject of the manuscript is very original, especially due to the fact that epigenetic mechanisms are involved in the regulation of all biological processes i.e., genome reorganization as well as cell differentiation. The authors describe the issues in a good manner, the manuscript is well composed. The results are clear and presented in an accessible and clear way for the reader. For that reason, I am in favor of publishing the manuscript in its current form.

Author Response

(The authors gave the same response as above.)

Reviewer 3 Report

Comments and Suggestions for Authors

In general:

The work shows how the inhibition of HDAC8 regulates the proliferation of non NSCs. The manuscript is of high quality and all data included are relevant and interesting. Epigentics is essential for neurogenesis. This can be taken for granted. It therefore makes sense to take a closer look at histones, as in this publication. The results obtained for HDAC8 appear interesting and plausible.

In terms of content:

I am most critical of the use of the inhibitor PCI-34051, which is hardly categorized in terms of method, content and function. It is also not explained how HDAC8 is inhibited. All results are based on the inhibitor, which is why this is a very important point for me. How can it be ruled out that the effect on HDAC8 is not simply an artifact or that HDAC8 is one of many targets (off targets)?

In my view, NES is one of many markers for neuronal progenitor cells, but it is not really specific for NSCs and therefore I am critical of the formulations that classify NES as an NSC marker. Other markers would also have been more appropriate from a developmental biology perspective. But is there a possible reason for the preference for NES?

How can the big difference between the KI67 measurements in the SVZ and the EdU measurement in the neurospheres be explained?

For the in vitro cultures, it is difficult for the reader to assess how pure and vital the isolated primary cells were and to what extent this may have affected the in vitro cultures. There is no classification.

Fig. 5 (a) The controls appear to show much greater variability than the treated group - please explain

Discussion:

The discussion spends a lot of time on the classification of HDAC8 and its relevance in different models - a bit like a review (shorten if necessary). Histone modification is briefly discussed in the context of deacetylation, but it is much more complex and also highly relevant to the results obtained. The classification of the results must be expanded here.

Furthermore, the RNAseq data and their evaluation are hardly discussed. How should the pathways obtained be classified? In my opinion, this is particularly interesting

In the introduction, the authors make a connection to tumorigenesis in which HDAC8 is expressed at an increased level (line 68). The authors themselves describe that the proliferation of NSCs is also associated with the expression of HDAC8. In my opinion, however, a classification of the possibly different cell types is missing here if the cited work is based on tumors from other cell types.

Figures:

Figure 2: Abbreviations "LV", "p.o." "SVG" not explained (explained later)

Figure 5: (b/g) explain "z-score" more preciciely (f) It is unclear to which data points the labeling should belong. It is also not clear why other data points are not labeled with higher values. (h) Asterisks and dots too small

Author Response

(The authors gave the same response as above.)
